# CD8+ Lymphocytes from Healthy Blood Donors Secrete Antiviral Levels of Interferon-Alpha

**DOI:** 10.3390/v15040894

**Published:** 2023-03-30

**Authors:** Fernando Teque, Abby Wegehaupt, Ellen Roufs, M. Scott Killian

**Affiliations:** 1Department of Medicine, University of California San Francisco, San Francisco, CA 94143, USA; 2Department of Basic Biomedical Sciences, Sanford School of Medicine, University of South Dakota, Vermillion, SD 57069, USA

**Keywords:** CD8+ T cells, cytokines, HIV, antiviral immune response, interferon alpha

## Abstract

The adaptive immune response to viral infections features the antigen-driven expansion of CD8+ T cells. These cells are widely recognized for their cytolytic activity that is mediated through the secretion of cytokines such as perforin and granzymes. Less appreciated is their ability to secrete soluble factors that restrict virus replication without killing the infected cells. In this study we measured the ability of primary anti-CD3/28-stimulated CD8+ T cells from healthy blood donors to secrete interferon-alpha. Supernatants collected from CD8+ T cell cultures were screened for their ability to suppress HIV-1 replication in vitro and their interferon-alpha concentrations were measured by ELISA. Interferon-alpha concentrations in the CD8+ T cell culture supernatants ranged from undetectable to 28.6 pg/mL. The anti-HIV-1 activity of the cell culture supernatants was observed to be dependent on the presence of interferon-alpha. Appreciable increases in the expression levels of type 1 interferon transcripts were observed following T cell receptor stimulation, suggesting that the secretion of interferon-alpha by CD8+ T cells is an antigen-driven response. In 42-plex cytokine assays, the cultures containing interferon-alpha were also found to contain elevated levels of GM-CSF, IL-10, IL-13, and TNF-alpha. Together, these results demonstrate that the secretion of anti-viral levels of interferon-alpha is a common function of CD8+ T cells. Furthermore, this CD8+ T cell function likely plays broader roles in health and disease.

## 1. Introduction

CD8+ T cells are a functionally and phenotypically heterogenous population of cells that comprise roughly 20–30% of circulating lymphocytes and play an important role in the host defense against viral infections [1,2]. CD8+ T cells have been extensively investigated in the context of human immunodeficiency virus-1 (HIV-1) infection and strong evidence links these cells with control of virus replication and protection from disease progression [3]. Notable findings are (i) circulating levels of simian immunodeficiency virus (SIV) rapidly increase following the in vivo-depletion of CD8+ T cells from SIV-infected macaques [4], (ii) features of CD8+ T cells and human leukocyte antigen (HLA) molecules differ between HIV-1 infected individuals having low and high viral loads [5,6,7], and (iii) characteristics of CD8+ T cells, independent of viral load, can predict the rate of HIV-1 disease progression [8].

CD8+ T cells from HIV-1-infected individuals were first reported to control HIV-1 replication in vitro via a non-cytolytic mechanism that involved the secretion of a soluble factor(s) [9]; reports of HIV-specific cytotoxic T cell responses soon followed [10]. To date, the relative contributions of non-cytotoxic and cytotoxic CD8+ T cell responses to the control of HIV replication have not been established. This unresolved issue has important implications for vaccines and therapeutic approaches designed to elicit strong CD8+ T cell responses, as it remains unclear whether the ideal approaches should elicit non-cytotoxic, cytotoxic CD8+ T cell responses or perhaps both. The need for improved understanding of the immunobiology of CD8+ T cells, including the cytokines that they secrete, is underscored by shortcomings of the STEP, HTVN505, and HTVN506 vaccines, designed to elicit strong HIV-specific T cell responses [11,12,13].

Focusing on non-cytotoxic T cell responses, several prominent research laboratories have attempted to identify soluble anti-HIV factors that are secreted by CD8+ T cells, including those led by Fauci [14], Ho [15], Gallo [16], and Levy [9]. Nonetheless, the lack of identification of the major anti-HIV factor secreted by CD8+ T cells has precluded its specific study in vitro and in vivo. In the lengthy search for the CD8+ T cell anti-HIV factor (CAF) first described by Walker et al. [9], several potential candidates have emerged, including β-chemokines (MIP-1α, MIP-1β, RANTES) [16]. Unlike β-chemokines, CAF is effective against both R5- and X4-tropic HIV-1 isolates [17], inhibits a post-entry stage in the virus life cycle [18], and correlates with clinical state [19]. 

In prior studies we have found that CD8+ T cells from select HIV-1-infected individuals secrete type 1 interferons (i.e., IFN-α,β,ω) and that the anti-HIV activity of the resulting CD8+ T cell culture supernatants is dependent on the presence of IFNα, strongly linking CAF with that type 1 interferon [20]. Noteworthy is that CD8+ T cells have long been recognized for their ability to secrete interferon gamma (INFγ; the sole type 2 interferon) [21], but unlike IFNα, IFNγ has no direct antiviral activity against HIV-1 in primary cell cultures or when administered therapeutically [22]. In contrast, little attention has been given to the production of IFNα by CD8+ T cells, despite the potent ability of IFNα to suppress HIV-1 replication in vitro and to reduce viral loads in treated patients [20,23,24,25] (see Table 1). To determine whether the secretion of antiviral levels of IFNα is a normal CD8+ T cell response or perhaps unique to certain HIV-1-infected individuals, in the present study we investigated primary CD8+ T cells obtained from healthy HIV-negative individuals.

## 2. Methods

*Blood donors*. Peripheral blood from healthy donors (N = 34) was procured from local blood banks. All blood donors signed informed consent forms and this study received institutional review board approvals from the University of California San Francisco and the University of South Dakota.

*Isolation of CD8+ and CD4+ T cells.* Leukocytes were isolated from peripheral blood by density-gradient separation over ficoll (GE) according to the manufacturer’s instructions [32]. CD4+ and CD8+ T cells were purified (n = 34 donors) using immunomagnetic beads (Miltenyi) as previously described [20]. Cell counts and viability assessments were performed using trypan dye exclusion and a standard hemocytometer. The resulting cell subsets were routinely ≥95% pure upon flow cytometric analysis. 

*CD8+ T cell culture supernatants*. Immediately following their isolation, purified CD8+ T cells were resuspended in growth medium (RPMI 1640, 10% FBS, 100 U/mL rhIL2 (Invitrogen) and antibiotics) and then were cultured as illustrated in Figure 1. Briefly, CD8+ T cells were stimulated with anti-CD3/CD28 beads (Invitrogen) according to the manufacturer’s protocol. After 3 days, the CD8+ cells (2 × 10^6^ cells/mL) were passed into complete serum-free medium (F12/DMEM, 1X ITS (Invitrogen), 100 U/mL rhIL-2, and antibiotics) and cultured for an additional 7–11 days. CD8+ T cells were passaged each 2–3 days to maintain a density of 2 million cells/mL in fresh serum-free medium. Supernatants were collected at each passage, centrifuged (1000× *g* for 15 min) to remove cellular debris, and stored at −80 °C. 

*HIV-inhibition assays*. The effects of CD8+ T cells directly, and of CD8+ T cell culture supernatants, on HIV-1-replication levels in heterologous primary human CD4+ T cells were measured as previously described [20,33,34]. Briefly, freshly isolated CD4+ T cells were stimulated with PHA for 3 days, acutely infected with a primary β-chemokine-insensitive HIV-1 isolate (X4 tropic; 10,000 TCID50) [35], and then cultured in the presence (experimental condition) or absence (control condition) of CD8+ T cells (1:1 input ratio) or CD8+ T cell fluids (50% *v*/*v*). On day 3 of the HIV-inhibition assay, 50% of the cell culture medium was collected and replaced with fresh medium or CD8+ T cell fluid. On day 6 of the HIV-inhibition assay, the cell culture supernatants were collected, centrifuged (1000× *g* for 15 min) to remove cellular debris, and HIV-1 p24 levels were measured by ELISA (NIH AIDS Reagents Program). Anti-HIV activity was calculated as:(1)$$\%\;\mathrm{suppression}=\left(1-\frac{\left[p24\;\mathrm{in}\;\mathrm{experimental}\;\mathrm{condition}\right]}{\left[p24\;\mathrm{in}\;\mathrm{control}\;\mathrm{condition}\right]}\right)\times 100\%$$

*Immunoassays*. CD8+ T cell culture supernatants were concentrated (>20×) using 10 kDa MWCO filters (Millipore). IFNα levels in the resulting concentrates were measured using an ELISA kit (PBL) that detects all human IFNα subtypes. Multiplex cytokine arrays (Millipore, 42-plex) were performed to measure the levels of cytokines and other soluble factors in concentrated CD8+ T cell culture supernatants (n = 6 unique fluids). The 42 proteins measured were EGF, Eotaxin-1, FGF-2, Flt-3L, Fractalkine, G-CSF, GM-CSF, GROα, IFNα2, IFNγ, IL-10, IL-12P40, IL-12P70, IL-13, IL-15, IL-17A, IL-18, IL-1α, IL-1B, IL-1RA, IL-2, IL-3, IL-4, IL-5, IL-6, IL-7, IL-8, IL-9, IP-10, MCP-1, MCP-3, MDC, MIP-1α, MIP-1γ, PDGF-AA, PDGF-BB, RANTES, sCD40L, TGFα, TNFα, TNFγ, and VEGF-A. The levels of IFNα and other cytokines reported here have been adjusted to account for the volumetric concentration of the fluids. 

*Flow cytometry.* Freshly isolated T cells were stained with CD4 PE and CD8 APC (BD Biosciences), washed, and resuspended in isotonic buffer. Flow cytometric analyses were performed using Accuri C6 and LSRFortessa cytometers (BD) and FlowJo v10 software (FlowJo).

*Measurement of interferon gene expression by qRT-PCR*. Transcript levels of *IFNA2*, *IFNA4*, *IFNA8*, *IFNA17*, *IFNB*, *IFNW*, and *IFNG* were measured by qRT-PCR in stimulated and resting CD8+ T cells (n = 3 donors). Briefly, RNA was extracted from cell lysates using an RNeasy mini kit (Qiagen, Hilden, Germany), including steps for homogenization (Qiashredder) and DNase digestion, as recommended by the manufacturer. RNA concentrations and purities (Abs 260:280) were measured with a NanoDrop ND-1000 spectrophotometer (Thermo Scientific). One-step qRT-PCR reactions were performed using a Quantitect Sybr green RT-PCR kit (Qiagen) as recommended by the manufacturer. 200 ng of RNA was added to each reaction along with interferon-specific primer pairs previously described [36]. Amplification and detection were performed using an ABI Prism 7700 Sequence Detection System (Applied Biosystems). Data were analyzed with SDS Software (Applied Biosystems; version 1.5.1) and then exported to Microsoft Excel. Fold-changes in mRNA transcript levels were computed using the 2-DC’T method [37].

*Data analysis.* Laboratory data were compiled in Excel spreadsheets (Microsoft). Data analyses and charting were performed using SigmaPlot v14.5 (Systat). For analysis of the multiplex cytokine arrays, the data were log2 transformed and normalized and principal component analysis (PCA) and heatmaps were generated using Qlucore Omics Explorer v3.6 (Qlucore). Statistical significance was set at the conventional level of α = 0.05.

## 3. Results

### 3.1. Antiviral Activity of CD8+ T Cells and CD8+ T Cell Fluids from Healthy Blood Donors

Previous studies have focused on the anti-HIV activities of CD8+ T cells and CD8+ T cell fluids from HIV-1 infected individuals [9,20,34,38]. In this study, similar approaches were used to determine whether CD8+ T cells and CD8+ T cell fluids from HIV-1 negative individuals can suppress HIV-1 replication (Figure 2). In direct cell contact assays with acutely HIV-1-infected CD4+ T cells (1:1 input ratio), the ability of CD8+ T cells to suppress HIV replication was observed to be dependent on exogenous stimulation (Figure 2A). Without stimulation, resting CD8+ T cells from healthy blood bank donors exhibited very little anti-HIV activity (generally < 20% suppression). Upon stimulation, the CD8+ T cell anti-HIV activity markedly increased to levels exceeding 50% suppression. Upon evaluation of CD8+ T cells from 8 healthy donors we observed that none of the unstimulated CD8+ T cells (n = 8) exhibited > 50% suppression, whereas 83% of the stimulated CD8+ T cells (n = 6) exhibited > 50% suppression when co-cultured with acutely infected CD4+ T cells (*p* < 0.01, Mann-Whitney U test). These results demonstrate that CD8+ T cells from healthy blood bank donors can suppress HIV-1 replication in a manner that is dependent upon the cell activation state. In addition, these results support the requirement of stimulation of CD8+ T cells from healthy blood donors prior to assessments of their secreted anti-HIV factors. 

Upon observing that stimulated CD8+ T cells from healthy blood bank donors can suppress HIV-1 replication in cell contact assays, we next evaluated the anti-HIV activity of conditioned medium (supernatants) derived from stimulated CD8+ T cell cultures of healthy donors. Select CD8+ T cell culture supernatants derived from most, but not all, of the blood bank donors exhibited appreciable anti-HIV activity, as evidenced by >50% suppression of HIV-1 replication in virus inhibition assays (Figure 2B). Longitudinal analysis of the relative antiviral activities of supernatants collected during the course of individual CD8+ T cell cultures revealed marked heterogeneity in the levels of anti-HIV activity over time, with peak activity being most commonly observed at days 5–6 of cell culture. For this study, we evaluated the anti-HIV-1 activity of CD8+ T cell culture supernatants derived from 31 healthy donors. Among the CD8+ T cell fluids evaluated at cell culture day 5, 21/30 (70%) exhibited the ability to suppress HIV-1 replication by ≥50%. The day 5 time point was the earliest evaluation of the CD8+ T cell culture supernatants and represents 48 h post-transfer of the cells to serum-free medium. In comparison, only 10% of the day 7 CD8+ T cell culture supernatants exhibited the ability to suppress HIV-1 replication by ≥50%. In this regard, CD8+ T cell culture supernatants from healthy blood bank donors exhibit kinetics similar to those previously observed with HIV-1-infected individuals [20]. This temporal variation in anti-HIV-1 activity enables the evaluation of soluble factors that are present in distinct CD8+ T cell culture fluids (e.g., those showing < or ≥ 50% suppression) derived from individual donors and between donors.

### 3.2. Interferon Transcript Levels in Resting and Stimulated CD8+ Cells

To investigate whether type 1 interferon genes are expressed in healthy donor CD8+ T cells, qRT-PCR assays were performed to measure the relative levels of mRNA transcripts of *IFNA2*, *IFNA4*, *IFNA8*, *IFNA17*, *IFNB*, and *IFNW* (Figure 3). In freshly isolated CD8+ T cells, type 1 interferon transcripts were detectable in low amounts as indicated by C_T_ > 25 (Figure 3A). Notably, IFNγ transcripts were also detected at similarly low levels. Among the type 1 interferons evaluated, transcripts for *IFNA2* and *IFNB* were the most abundant in freshly isolated CD8+ T cells. Following T cell receptor stimulation, the same CD8+ T cells exhibited appreciable increases in the levels of all evaluated type 1 interferon transcripts, albeit much lower than the observed increase in IFNγ transcripts. Comparison of the levels of type 1 interferon transcripts present in resting and stimulated CD8+ T cells revealed a predictable pattern of differential changes (Figure 3B). The largest increase occurred in *IFNA8*, followed by *IFNW, IFNB,* and *IFNA17*. Overall, the increased levels of type 1 interferon transcripts that were observed following stimulation of the CD8+ T cells were greater than would be expected by chance (Paired *t*-test; *p* = < 0.01). These results demonstrate that CD8+ T cells from healthy individuals express type 1 interferon transcripts and that their levels are increased upon T cell receptor stimulation. In addition, our results extend previous findings that T cells contain *IFNA* mRNA transcripts both in the absence of an inducer and following induction with Sendai virus [39].

### 3.3. IFNα Levels in CD8+ T Cell Fluids from Healthy Blood Donors

To determine whether not the observed antiviral activity of CD8+ T cell fluids from healthy blood donors was attributable to the presence of IFNα, ELISAs were performed for 36 supernatants collected at distinct time points from CD8+ T cell cultures established from 19 donors (Figure 4). Temporal analysis revealed a direct association between anti-HIV-1 activity and IFNα levels for individual CD8+ T cell cultures (Figure 4A). Thus, the chronological variation in the anti-HIV-1 activity of CD8+ T cell culture supernatants collected from individual cultures can be explained by the variable in presence of IFNα in the cultures. IFNα was detectable and present at significantly higher levels (*p* < 0.001) in fluids having > 50% anti-HIV-1 activity (Figure 4B). In fact, all of the CD8+ T cell culture supernatants that exhibited > 50% suppression of HIV contained detectable IFNα. After adjustment of the measured IFNα level with respect to the volumetric concentration factor, the CD8+ T cell culture supernatants exhibiting > 50% suppression of HIV-1 (n = 13) had a median IFNα concentration of 6.5 pg/mL (range = 1.2–28.6). In comparison, the fluids exhibiting < 50% suppression (n = 23) had a median IFNα concentration of 0.1 pg/mL (range = 0–1.5). Concentrated cell culture medium alone did not have any detectable IFNα (data not shown). A direct correlation was observed between the measured anti-HIV-1 activity of the fluids and the concentration of IFNα in the fluids (r^2^ = 0.729; Figure 4C).

In addition, we performed a receiver operating characteristic (ROC) analysis to assess the ability of the ELISA-measured IFNα concentration to classify the anti-HIV-1 activity (<50% or >50%) of the CD8+ T cell culture supernatant (Figure 4D). The resulting area under the ROC curve was strikingly high (AUC = 0.993, *p* < 0.001). IFNα concentrations of 0.5 pg/mL and 3.9 pg/mL bridged the boundaries between 100% specificity and 100% sensitivity values respectively. The ROC analysis indicated that an IFNα concentration threshold of 0.8 pg/mL can predict the anti-HIV-1 activity status with 96% specificity and 100% sensitivity. In previous studies we have shown that approximately 1 pg/mL of IFNα can cause a 50% reduction in HIV-1 replication in our assay [20]. Therefore, these results demonstrate that CD8+ T cells from healthy blood donors secrete IFNα at levels that can suppress HIV-1 replication in vitro.

### 3.4. Cytokine Profiles of CD8+ T Cell Fluids from Healthy Blood Donors

Milliplex 42-plex cytokine assays (Millipore) were used to analyze and compare the cytokine profiles of CD8+ T cell fluids with (n = 3; >50% suppression) and without (n = 3; <50% suppression) anti-HIV-1 activity (Figure 5). Median concentration values were evaluated to establish the relative abundances of each soluble factor present in the fluids (Figure 5A). Substantial variation (~4 logs) was observed among the median concentrations of the 42 analytes measured. The levels of Eotaxin-1, IL-15, IL-7, IL-12P70, G-CSF, IL-1B, IL-12P40, IL-18, EGF, sCD40L, TGFα, MCP-3, and IL-17A were consistently undetectable/very low (<1 pg/mL) and did not vary in magnitude with the anti-HIV-1 activity of the fluids. Thus, these analytes can be reasonably excluded as anti-HIV-1 factors secreted by CD8+ T cells. Also observed was that the b-chemokines MIP-1a (CCL3), MIP-1b (CCL4) and RANTES (CCL5) were among the most abundant proteins measured. While IL-2 was observed to be present in the fluids at a relatively high concentration, it should be noted that this cytokine was added exogenously to the cultures to promote CD8+ T cell growth. 

To identify a potential composite cytokine signature that distinguishes CD8+ T cell culture supernatants having anti-HIV activity > 50%, a two-group covariance-based principal component analysis (PCA) was performed on transformed (log2) and normalized (mean = 0, variance = 1) data (Figure 5B). Among the 42 analytes included in the PCA, 5 cytokines were identified when the significance threshold was restricted to *p* = 0.05 (q = 0.276). Validating our ELISA findings, IFNα was present at appreciable levels in the CD8+ T cell culture supernatants having anti-HIV-1 activity (>50% suppression) and not in supernatants lacking this activity (*p* < 0.001). Concomitant with elevated IFNα levels, the CD8+ T cell culture supernatants having anti-HIV-1 activity had elevated levels of GM-CSF, IL-10, IL-13, and TNFα (Figure 5C). None of the 42 analytes measured were appreciably decreased in the supernatants having >50% anti-HIV-1 activity.

## 4. Discussion

To our knowledge, this is the first report that primary CD8+ T cells from healthy blood donors can secrete antiviral levels of IFNα. It has previously been reported that various lymphoblastic T cell lines, including the CD3+CD8+TCRα/β+ HPB-ALL cell line, can secrete appreciable levels of IFNα and IFNβ [40,41]. In addition, by immunofluorescent microscopy, it has been shown that nearly all primary T cells from healthy blood donors constitutively express intracellular IFNα [42]. The present discovery was facilitated by our novel cell culture procedure that derives CD8+ T cell culture supernatants from serum-free medium (Figure 1). Consequently, the supernatants can be concentrated using low molecular weight cut-off (MWCO) filters allowing for the detection and measurement of low-level proteins (e.g., <50 pg/mL). In contrast, fluids containing serum often clog low MWCO filters, such as 10kDa filters that are needed to concentrate cytokines that are generally <40 kDa.

Our studies show that primary CD8+ T cells from healthy blood donors can inhibit HIV-1 replication in vitro in cell:cell contact assays (Figure 2a). While this CD8+ T cell antiviral activity required T cell receptor stimulation, it is reasonably not HIV-specific because the healthy blood bank donors were not likely to have been exposed to HIV. We and others have previously shown that the anti-HIV-1 activity of CD8+ T cells from HIV-1-infected individuals is mediated by cells having a memory phenotype (i.e., CD45RA-, CCR7-, CD27+, PD1+ cells) [33,43]. In comparison to prior studies of CD8+ T cells from asymptomatic HIV-1-infected individuals [33], CD8+ T cells from healthy blood bank donors appear to exhibit reduced anti-HIV-1 activity. This observation can be explained by the relatively reduced frequency of memory CD8+ T cells in healthy blood bank donors. Nonetheless, additional studies are required to determine if the anti-HIV-1 activity of CD8+ T cells from healthy donors is mediated by memory cells.

Importantly, the present studies demonstrate that CD8+ T cell culture supernatants from healthy blood bank donors can also suppress HIV-1 replication (Figure 2b). While most of the individual CD8+ T cell cultures yielded at least one supernatant having >50% anti-HIV-1 activity, the timing of those cell culture supernatants was variable. This observation is perhaps due to donor-donor variation in the frequency of memory CD8+ T cells and/or variation in the growth rates of the cells. In comparison to prior results [20,44], the CD8+ T cell culture supernatants derived from healthy blood donors and asymptomatic HIV-1-infected individuals exhibit roughly equivalent levels of anti-HIV activity (generally < 75% suppression).

The major goal of this study was to determine whether CD8+ T cells from healthy blood donors secrete anti-viral levels of IFNα as a general immune response. Indeed, we found that IFNα transcripts levels become elevated following the T cell receptor stimulation of CD8+ T cells (Figure 3) and that IFNα is present in cell culture supernatants derived from the stimulated CD8+ T cells of healthy blood donors (Figure 4). Temporal heterogeneity in the anti-HIV-1 activity of individual CD8+ T cell cultures was associated with the presence or absence of IFNα (Figure 4a). Furthermore, the anti-HIV activity of the supernatants was observed to be dependent on the presence of IFNα (Figure 4b). The IFNα concentration that was measured in the supernatants having anti-HIV-1 activity is consistent with the IFNα concentration required for the observed level of suppression [20]. Also, in previous studies of CD8+ T cell culture supernatants derived from HIV-1-infected individuals, we observed that neutralization of type 1 interferon or blocking of the type 1 interferon receptor abrogated the majority of the anti-HIV activity of those fluids [20]. And, well-established antiviral functions of IFNα [45] are congruent with studies that have identified disruption of viral gene expression as the mechanism of action of CD8+ T cell culture supernatants having anti-HIV-1 activity [46,47,48,49]. These observations strongly implicate IFNα as being the major anti-HIV-1 factor secreted by CD8+ T cells.

Here, we have also provided an assessment of the concentrations of 42 soluble factors that can be present in CD8+ T cell culture supernatants (Figure 5). In addition to IFNα, we found GM-CSF, IL-10, IL-13, and TNFα to be present at elevated levels in CD8+ T cell culture fludis having anti-HIV-1 activity. GM-CSF is known to be produced by activated T cells [50], but it does not inhibit HIV-1 replication in CD4+ T cells [51] and its receptor (GMCSFR/CD116) is not expressed on normal T cells [52]. IL-10 does not inhibit HIV-1 replication in T cells and does not act to inhibit HIV-1 replication at the transcriptional level [53]. Moreover, IL-13 [54] and TNFα [55] do inhibit HIV-1 replication in T cells. 

Nonetheless, it is interesting to speculate that the co-secretion of IFNα with these additional cytokines characterizes a novel population of polyfunctional CD8+ T cells.

The secretion of IFNα by CD8+ T cells can have broad implications in HIV-1 infection and other diseases. While all 13 subtypes of IFNα and the other type 1 interferons commonly bind and signal through the type 1 interferon receptor (IFNAR) [56], their effects are context-specific and can be beneficial or harmful [57]. In the context of HIV-1 infection, IFNα can act to restrict virus replication, promote viral latency, contribute to chronic inflammation, and/or exert cytostatic effects leading to the gradual decline of CD4+ T cells. These potential effects of IFNα-secreting CD8+ T cells merit consideration in HIV vaccine design and cure strategies. In the context of autoimmune diseases having evidence of a type 1 interferon gene signature, such as lupus, Sjögren’s syndrome, systemic sclerosis and rheumatoid arthritis [57], IFNα-secreting CD8+ T cells could be important mediators of the sequelae. And in the context of infections and cancer, subtleties in the timing, specificity, and magnitude of the IFNα-secreting CD8+ T cell response could contribute to variation in clinical outcomes.

In summary, secretion of IFNα appears to be a fundamental CD8+ T cell response. Given that this activity can be attributed to memory cells, the secretion of IFNα by CD8+ T cells is likely a previously unrecognized adaptive immune response. Our findings warrant further investigation of the roles of IFNα-secreting CD8+ T cells in health and disease.

## Figures and Tables

**Figure 1 viruses-15-00894-f001:**
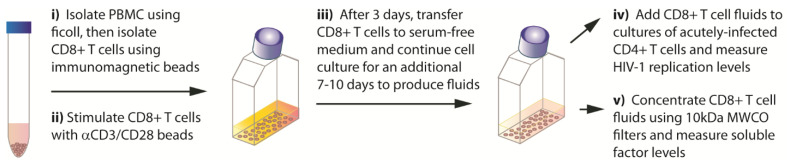
Overview of the procedures used to produce and evaluate CD8+ T cell culture supernatants.

**Figure 2 viruses-15-00894-f002:**
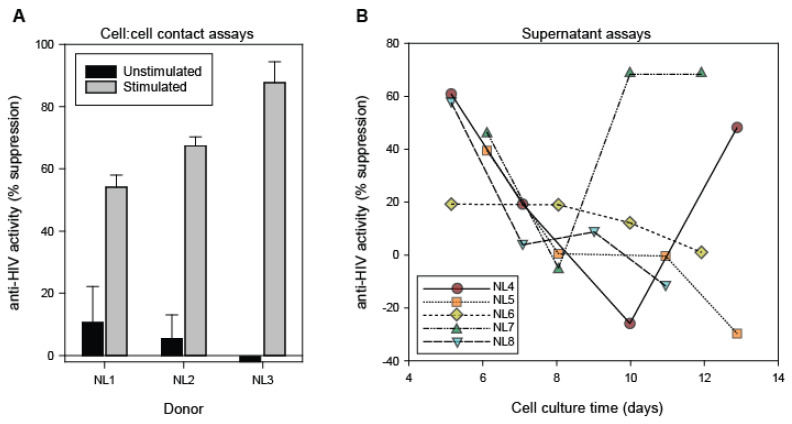
*Antiviral activity of CD8+ T cells from healthy blood bank donors*. (**A**) Freshly isolated CD8+ T cells (n = 8 donors) were cultured for 3 days in the absence (unstimulated) or presence (stimulated) of anti-CD3/CD28 beads and then added to cultures of acutely HIV-1-infected CD4+ T cells. Shown are results of the CD8+ T cells from 3 representative healthy blood bank donors to suppress HIV-1 replication in CD4+ T cells in cell:cell contact assays. (**B**) Supernatants collected from cultures of stimulated CD8+ T cells were evaluated for anti-HIV-1 activity (n = 31 donors). Results are shown for the anti-HIV-1 activities of CD8+ T cell culture supernatants derived from 5 representative healthy blood bank donors and measured for each cell passage. The temporal CD8+ T cell culture supernatants derived from an individual donor were assayed simultaneously using common acutely HIV-infected CD4+ T cells, whereas the source of the CD4+ T cells varied among assessments of CD8+ T cell culture supernatants from different donors.

**Figure 3 viruses-15-00894-f003:**
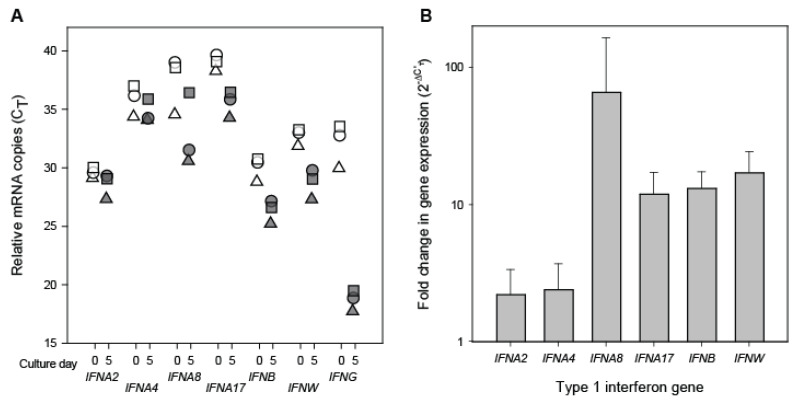
*Interferon transcript levels in resting and stimulated CD8+ cells*. Relative levels of type 1 interferon (*IFNA2*, *IFNA4*, *IFNA8*, *IFNA17*, *IFNB*, and *IFNW*) and type 2 interferon (IFNG) mRNA transcripts were measured by qRT-PCR. (**A**) Shown are the C_T_ values for each transcript measured in resting (open symbols) and stimulated (shaded symbols) CD8+ T cells from 3 different donors. (**B**) Shown are fold-changes (means and standard deviations) in the expression of type 1 interferon transcripts upon stimulation of the CD8+ T cells and measured at day 5 of the cell culture.

**Figure 4 viruses-15-00894-f004:**
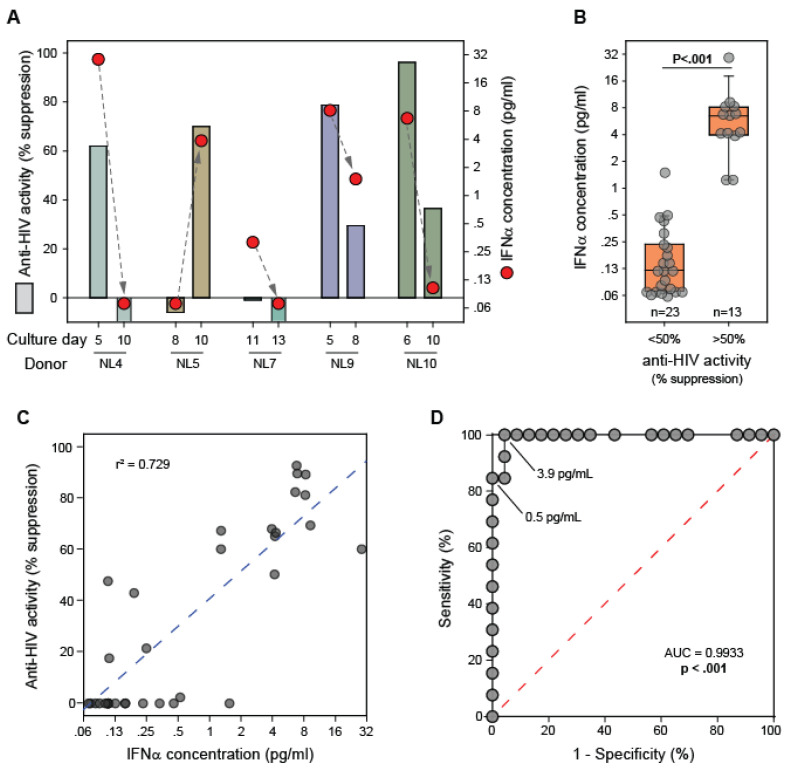
*IFNα levels and anti-HIV-1 activities of CD8+ T cell culture supernatants*. Supernatants were collected from CD8+ T cell cultures (n = 19 donors) at distinct time points and evaluated for anti-HIV-1 activity and IFNα concentration. (**A**) Longitudinal results are shown for the measured anti-HIV-1 activity and corresponding IFNα concentration for CD8+ T cell culture supernatants derived from 5 representative healthy blood bank donors. (**B**) Comparisons of the IFNα levels between CD8+ T cell culture supernatants having <50% and >50% anti-HIV-1 activity. Boxes denote medians and interquartile ranges, and whiskers denote 10th and 90th percentiles for each group. (**C**) Shown is the correlation between the anti-HIV-1 activity and the IFNα concentration measured in 36 unique CD8+ T cell culture supernatants. (**D**) Shown is a receiver operating characteristic (ROC) curve that evaluates the ability of the ELISA-measured IFNα concentration to classify the anti-HIV-1 activity (<50% or >50%) of the CD8+ T cell culture supernatant. Noted are IFNα concentrations that bridge 100% specificity and 100% sensitivity values. Area under the curve (AUC).

**Figure 5 viruses-15-00894-f005:**
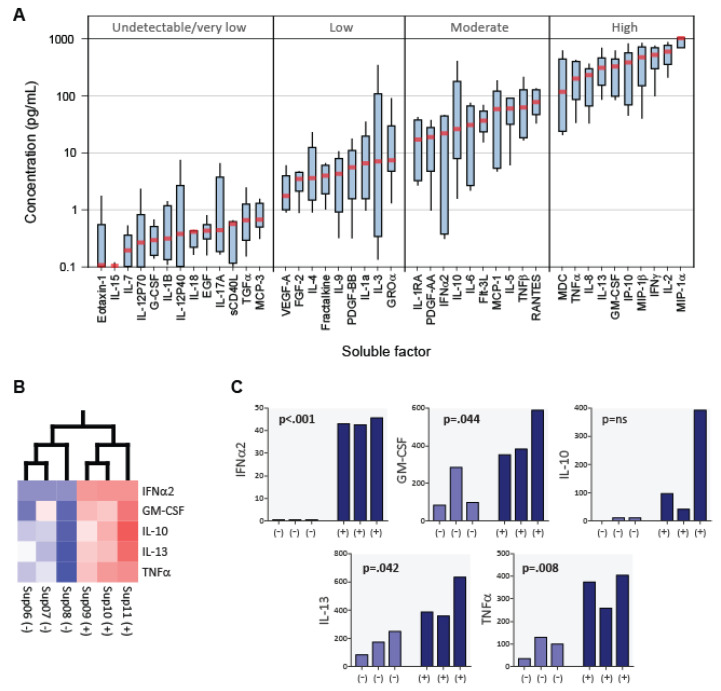
*Cytokines in CD8+ T cell culture supernatants*. Multiplex cytokine assays were performed on CD8+ T cell culture supernatants having <50% (−, n = 3) and >50% (+, n = 3) anti-HIV-1 activity. (**A**) Boxplots are shown for the 42 soluble factors measured, ranked in order of increasing median concentration. Boxes denote medians (red lines) and interquartile ranges, and whiskers denote 10th and 90th percentiles for analyte. (**B**) Among the 42 soluble factors evaluated, principal component analysis (PCA) identified 5 cytokines that distinguished the 2 groups. (**C**) Bar plots show the concentrations (pg/mL) of the distinguishing cytokines for each CD8+ T cell culture supernatant.

**Table 1 viruses-15-00894-t001:** Comparison of salient HIV-associated characteristics of type 1 and type 2 interferon.

Characteristic	Type 1 Interferon (IFNα, β, ι)	Type 2 Interferon (IFNγ)	References
Secreted by CD8+ T cells from healthy HIV-infected subjects	+	+	[20,22]
Associated with HIV-1 disease progression	+	+	[22,26]
Associated with prevention of HIV infection	+	+	[22,27]
Affects CD4+ T cell growth in vitro	−	−	[28,29]
Affects HIV transcription	+	−	[27]
Affects all HIV isolates tested	+	−	[20,30,31]

## Data Availability

All required data are available as figures in the main text.

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
