# Peer review of "CD8+ Lymphocytes from Healthy Blood Donors Secrete Antiviral Levels of Interferon-Alpha"

_viruses, 2023, doi:10.3390/v15040894_

Round 1

Reviewer 1 Report

The objective of this study is to evaluate the capacity of CD8 T cells to produce anti-viral activity by producing type I IFN (anti-viral IFN). The study is rather well constructed but it lacks statistical evaluation, and also phenotypic and functional characterization of the activated cells. However, it answers the problem in a partial way, but requires a more thorough study in the future in order to better realize this characterization. 

3.1 and Figure 2 : It would have been interesting to characterize the phenotype of the patients' CD8 T cells, in order to be able to present their activation (by flow cytometry with markers such as CD69, CD38 and HLADR) and differentiation states (by flow cytometry with markers such as CD45RA, CCR7, CD27).

Why do we find such heterogeneous anti-HIV activities in the supernatants? Has further research been done? MHC, phenotypic, activation studies? see previous comment. 

The number of samples seems to me rather low, between 3 healthy donors for anti-viral activity by cell contact and 5 for anti-viral activity linked to the supernatant. The absence of statistics, although the results are very clear, does not help to assure these results. 

Furthermore, why were CD8 T cells from HIV+ patients not used as an additional test? 

3.2 and fig 3: Again, the number of 3-donor samples seems low and having a few more donors would allow to confirm these results and to implement statistics that are missing. 

3.4 and figure 4: The evaluation of other cytokines in the different supernatants in relation to the inhibition of viral replication is clearly demonstrated. Nevertheless, it would be interesting to show that the 2 other cytokines (IL-13 and TNF-a) known to have antiviral activity are blocked to confirm the effect of type I IFN in this context. Nevertheless, the production of type I IFN by CD8 T cells remains a novel and interesting point, which needs to be confirmed, as explained by the authors in complementary studies, such as ICS (Intracellular Cytokines staining). 

The study seems convincing to me but has some shortcomings in the number of samples, statistical tests and phenotypic characterization of CD8 T cells. 

Reviewer 2 Report

CD8+ T cells are critical for controlling HIV-1 replication; however, the functional mechanism is poorly understood. In this study, the authors measured the ability of primary anti-CD3/28-stimulated CD8+ T cells from healthy blood donors to secrete interferon-alpha and further evaluated the association between interferon-alpha production and anti-HIV effects in CD8+ T cells. The study is well designed and presented. However, the data are not convincing due to the small sample size, in vitro, and lack of some essential experimental details. My concerns are listed below.

(1)    Only three donors were included in the analysis, thus, there were only tendency descriptions but no statistically analyzed results throughout the manuscript. We can hardly reach convincing conclusions only if more samples were analyzed with statistical significance.

(2)    Essential information regarding experimental procedures or quality controls was lacking. For example,

a.      in the co-culture experiment for HIV inhibition assays, were the CD8+ T cells and CD4+ T cells purified from the same donor? If so, how to maintain CD4+ T cells considering that CD8+ T cells experienced a more extended period of culture than CD4+ T cells?

b.      The efficiency of HIV-1 infection for each donor should be provided as supplemental data in the HIV inhibition assays.

c.      42 Multiplex cytokine arrays were performed to measure the levels of soluble factors in the supernatants. However, only GM-CSF, IL-10, IL-13, and TNF-a were mentioned. All the parameters should be mentioned in the method section and the PCA heatmap in Figure 5A.

(3)    HIV-inhibition assays with IFN blocking are needed to conclude that “The anti-HIV-1 activity of the cell culture supernatants was observed to be dependent on the presence of interferon-alpha.”

(4)    Figure 4B, was there a significant correlation between IFNa levels and anti-HIV activity?

Reviewer 3 Report

The title of this manuscript states that 'CD8+ T lymphocytes from healthy blood donors secrete antiviral levels of IFN-alpha'. Indeed the authors show that CD8+T cells from 'most, but not all' healthy donors stimulated with anti-CD3 and anti-CD28-antibody linked beads suppress HIV replication in X4 HIV-1-infected, CD4+ PHA blasts, and that they secrete IFN-alpha at pg/ml levels. The number of individuals who were tested is not stated in methods or results, but fig 4 shows that viral suppression to > 50% was found in 13 donors and not in 23 donors, and that this was significantly associated with the level of IFN-alpha found in the supernatants. This is a correlation. All cells produce some IFN-alpha after adequate stimulation. The result does not show that this IFN-alpha is responsible for the antiviral activity of the CD8+ T cell supernatants. To show this, treatment with anti-IFN-alpha antibodies should be used. It is however likely that IFN-alpha is not solely responsible for the antiviral activity, as indeed written in the discussion. This study is well-written but lacks some important features like a clear definition of the numbers of samples studied for each experiment, and I am not sure it brings any novelty or interesting confirmation to the field.

Round 2

Reviewer 2 Report

The authors have satisfactorily addressed all my concerns.

Author Response

Thank you for your review of our manuscript.

Reviewer 3 Report

Now that the study was explained more thoroughly and all of the results were shown, its interest appears more clearly.

The correlation between antiviral activity and IFN seems to be made only with type I IFNs and not with IFN-gamma. However, the expression of IFN-gamma mRNA changes a lot after stimulation. Can the authors add this correlation, or lack of correlation, please?  It is possible that blockade of IFN-a has an impact on IFN-gamma production, which also has an impact on HIV replication, at least in vitro. Indeed, it was shown that type I IFNs induce IFN-gamma intracellular production in CD8 T cell clones (Isnard, J Immunol 2021,, 207: 15-22, doi: 10.4049/jimmunol.2000392), which is released only upon cognate interaction. Even if the authors cite an article supporting  that IFN-gamma has no direct antiviral activity against HIV-1 in primary cell cultures or when administered therapeutically [22]), this might be added to enrich the discussion.

Author Response

Thank you for this comment.  We observed that IFNg was present at high levels in the CD8+ T cell culture fluids (Fig. 5A) and was not predictive of the anti-HIV activity.  The question of whether composite cytokines secreted by CD8+ T cells help to establish an antiviral state in the CD4+ T cells is interesting and warrants investigation in future studies.